

# Drone images afford more detections of marine wildlife than real-time observers during simultaneous large-scale surveys

Amanda J. Hodgson[1,2], Nat Kelly[3] and David Peel[4]

[1] School of Science, Edith Cowan University, Joondalup, Western Australia, Australia
[2] Harry Butler Institute, Murdoch University, Murdoch, Western Australia, Australia
[3] Australian Antarctic Division, Kingston, Tasmania, Australia
[4] Data 61, CSIRO, Hobart, Tasmania, Australia

Corresponding author
Amanda J. Hodgson,
a.hodgson@ecu.edu.au

## ABSTRACT

There are many advantages to transitioning from conducting marine wildlife surveys *via* human observers onboard light-aircraft, to capturing aerial imagery using drones. However, it is important to maintain the validity of long-term data series whilst transitioning from observer to imagery surveys. We need to understand how the detection rates of target species in images compare to those collected from observers in piloted aircraft, and the factors influencing detection rates from each platform. We conducted trial *ScanEagle* drone surveys of dugongs in Shark Bay, Western Australia, covering the full extent of the drone's range (∼100 km), concurrently with observer surveys, with the drone flying above or just behind the piloted aircraft. We aimed to test the assumption that drone imagery could provide comparable detection rates of dugongs to human observers when influenced by same environmental conditions. Overall, the dugong sighting rate (*i.e.*, count of individual dugongs) was 1.3 (95% CI [0.98–1.84]) times higher from the drone images than from the observers. The group sighting rate was similar for the two platforms, however the group sizes detected within the drone images were significantly larger than those recorded by the observers, which explained the overall difference in sighting rates. Cloud cover appeared to be the only covariate affecting the two platforms differently; the incidence of cloud cover resulted in smaller group sizes being detected by both platforms, but the observer group sizes dropped much more dramatically (by 71% (95% CI [31–88]) compared to no cloud) than the group sizes detected in the drone images (14% (95% CI [−28–57])). Water visibility and the Beaufort sea state also affected dugong counts and group sizes, but in the same way for both platforms. This is the first direct simultaneous comparison between sightings from observers in piloted aircraft and a drone and demonstrates the potential for drone surveys over a large spatial-scale.

# INTRODUCTION

Aerial surveys provide data on species distribution, abundance and habitat use that is critical for understanding the status of, and managing human impacts on, many species at risk.
For some species, aerial surveys are the only feasible method of undertaking assessments of distribution and abundance, including species that occur over large ranges and/or in remote locations, are more likely to be sighted from the air (*e.g.*, through water or vegetation), or are less likely to respond to aircraft than ground-based approaches (*Buckland et al., 2001*).

However, aerial surveys that involve observers on board light-aircraft spotting animals in real-time (observer surveys) involve significant risk to researchers (*Sasse, 2003*; *Hodgson, Kelly & Peel, 2013*), are often prohibitively expensive, are logistically challenging, and rely very heavily on the skill and experience of the observers. The potential advantages of conducting imagery surveys using drones (unoccupied aerial vehicles or UAVs) to survey wildlife has been widely acknowledged: they all-but eliminate human risk; have the potential to reduce survey costs; provide a permanent record of sightings, so allow validation of species and group sizes; provide comparatively more detailed and more accurate spatial data; offer flexibility to survey in remote locations; and significantly reduce the carbon footprint of surveys (*Hodgson, Kelly & Peel, 2013*; *Christie et al., 2016*; *Hodgson et al., 2016*).

It is now widely accepted that aerial imagery from drones offers a lot of exciting possibilities for surveys of wildlife (*Anderson & Gaston, 2013*; *Linchant et al., 2015*; *Gonzalez et al., 2016*). However, it is important to maintain the validity of long-term data series whilst transitioning from observer to imagery surveys. Long-term studies that involve regular surveys play an important role in understanding complex ecosystems, quantifying responses to environmental change, and providing data to support decision making and management of human activities (*Lindenmayer et al., 2012*). Therefore, we need to understand how the data collected in images from drones compare to those collected from observers in piloted aircraft. That is, the detection rates, and factors influencing detection rates, must be understood. There is no guarantee that detection rates from imagery will be as high as those from observers, however, imagery surveys may still be feasible if a minimum detection rate can be achieved, or the limitations of detection are understood (*Chabot, Craik & Bird, 2015*; *Van Andel et al., 2015*). Alternatively, if detection rates are improved by using drones, then an understanding of the mechanisms of this improvement is critical in order to compare results from observer and imagery survey data in long-term monitoring programs (*Hodgson et al., 2016*; *Hodgson, Peel & Kelly, 2017*).

In surveys of marine fauna, the detection of animals is imperfect—a certain proportion of animals will be too deep in the water column to be detected from the air at the moment the aircraft passes over them. This proportion must be accounted for if attempting to obtain a population abundance estimate. It is commonly understood that the probability of detecting an animal is affected by two factors: (1) availability probability—the proportion of time the animal is visible from the air, and (2) perception probability—the probability of an observer actually seeing the animal if it is available (*Marsh & Sinclair, 1989a*; Fig. 1). Availability has two components: the diving behaviour of the animal or group, and the environmental conditions such as water turbidity (*Pollock et al., 2006*). Diving behaviour can be affected by numerous factors such as water depth (*Hagihara et al., 2018*), group composition (*e.g.*, whether a calf is present) (*Hodgson, Peel & Kelly, 2017*), and behaviour (*Dorsey, Richardson & Würsig, 1989*). Various methods have been used and experiments
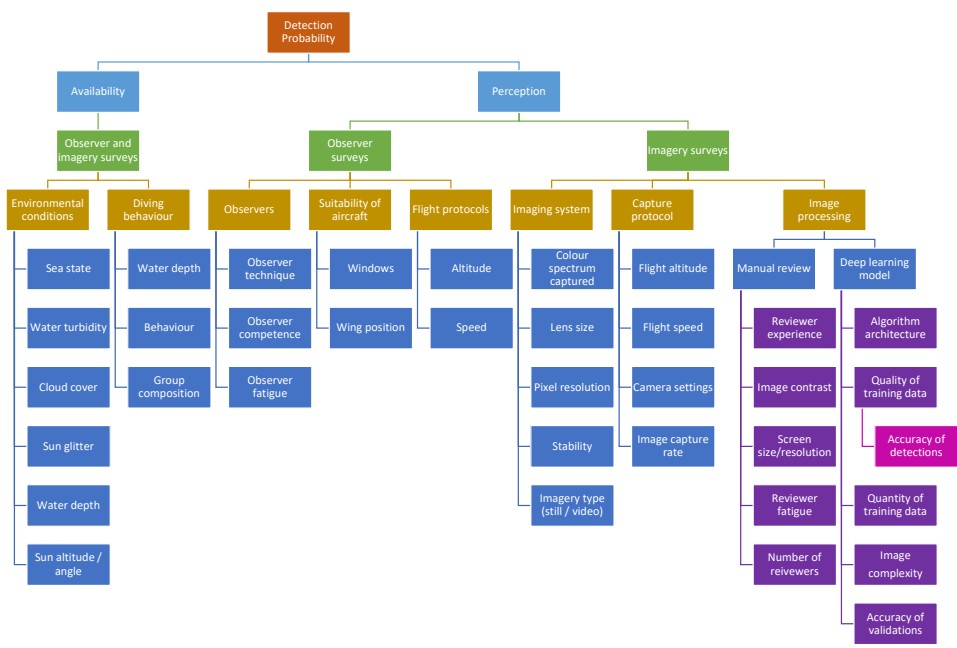

**Figure 1** **Factors that could potentially affect detection probability in marine mammal aerial surveys.**

conducted to estimate the availability of marine mammals (summarised in *Hodgson, Peel & Kelly (2017)*, see also *Hagihara et al. (2018)* and *Sucunza et al. (2018)* for examples). Converting from traditional observer surveys to drone surveys requires on understanding of whether using imagery rather than the human eye affects the availability of marine fauna. Drones offer an alternative method for assessing availability as this technology allows us to follow and observe the behaviour of marine fauna and directly observe the proportion of time individual animals, or groups, are available to be seen from the air. This method was demonstrated by *Hodgson, Peel & Kelly (2017)* using the *ScanEagle* drone to follow humpback whale groups, and has since been used to assess the availability of Australian humpback dolphins (*Sousa sahulensis*) (*Brown et al., 2022*). This technique may not be applicable to all species but does have a number of advantages over previously used methods, including that it assesses the availability from the same platform as that being used for the surveys (*Hodgson, Peel & Kelly, 2017*).

Methods to account for perception bias include using two independent observers on each side of the aircraft and implementing a mark-recapture model (*Buckland et al., 2004*; *Pollock et al., 2006*). Animals could also be missed when manually reviewing drone images to record sightings (Fig. 1), so similar methods to estimate perception bias need to be employed when using aerial imagery—an issue that has not yet been addressed in the literature.

Dugongs (*Dugong dugon*), which are regularly surveyed in a number of countries *via* large-scale strip-transect aerial surveys (*Marsh, O'Shea & Reynolds III, 2011*; *Cleguer et al., 2015*), present a particular challenge in that they mostly occur in shallow coastal areas where seagrass is found. Therefore, they can often be seen through the shallow water while

foraging on the sea floor. They can also be found in group sizes ranging from one to hundreds of animals. The 'availability' of dugongs during observer surveys has been well researched (*Pollock et al., 2006*; *Hagihara et al., 2014*; *Hagihara et al., 2018*). *Hodgson, Kelly & Peel (2013)* demonstrated that dugongs could be reliably detected in still images from the *ScanEagle* drone, and that detection may not be affected by sun glitter on the images, or sea state. However, these tests could not determine how detection rates compared to human observers, and therefore, the study was not sufficient to allow the transition of regular dugong surveys to drones.

An additional limitation of the previous drone dugong survey trial was that researchers were only permitted to fly the *ScanEagle* within line of sight. Legislation and permitting restrictions have been a major limitation to the adoption of drones for large-scale (100's km$^2$) wildlife surveys in many countries (*Christie et al., 2016*). This situation is still relatively dynamic as governing agencies develop new protocols and procedures for use of drones, with some countries relaxing restrictions while others are tightening them.

In our study, we were permitted to conduct trial *ScanEagle* surveys of dugongs in Shark Bay, Western Australia, covering the full extent of the drone's range (~100 km). To test the assumption that drone imagery could provide comparable detection rates of dugongs to human observers on-board aircraft, we flew both platforms over the same survey flight path at the same time, and were therefore able to compare detection rates for the same environmental conditions. We show that dugong sighting rates from the drone were superior to observer sighting rates, in that, we detected larger dugong group sizes in the drone images. This is the first direct simultaneous comparison between sightings from observers in piloted aircraft and a drone, and demonstrates the potential for drone wildlife surveys over a large spatial scale.

## MATERIALS & METHODS

### Study site and survey design

We conducted trial surveys in Shark Bay, Western Australia (25°30′S, 113°30′E), a large bay (13,000 km$^2$) divided into two embayments by the Peron Peninsula. The bay supports a high density of dugongs (~10,000 (*Marsh et al., 1994*; *Preen et al., 1997*; *Gales et al., 2004*; *Holley, Lawler & Gales, 2006*; *Hodgson, 2007*)) and therefore provided an ideal location to compare our two platforms.

The survey design followed that conducted in Shark Bay during five previous aerial surveys, which consists of a series of parallel line transects spaced 4.6 km apart (*Marsh et al., 1994*; *Preen et al., 1997*; *Gales et al., 2004*; *Holley, Lawler & Gales, 2006*; *Hodgson, 2007*) (Fig. 2). Unlike these previous surveys, we did not survey the entire bay, as the primary objective of this field trial was to compare the two survey methods. We therefore prioritised flying transects covering areas where the highest densities of dugongs were expected to occur to maximise our sample of dugong sightings. We chose to fly three of the blocks previously described and in block 4, we repeated some sections (Fig. 3).

Animal ethics approval to conduct the survey flights described below was obtained from the Murdoch University Animal Ethics Committee (permit number R2365/10). This work

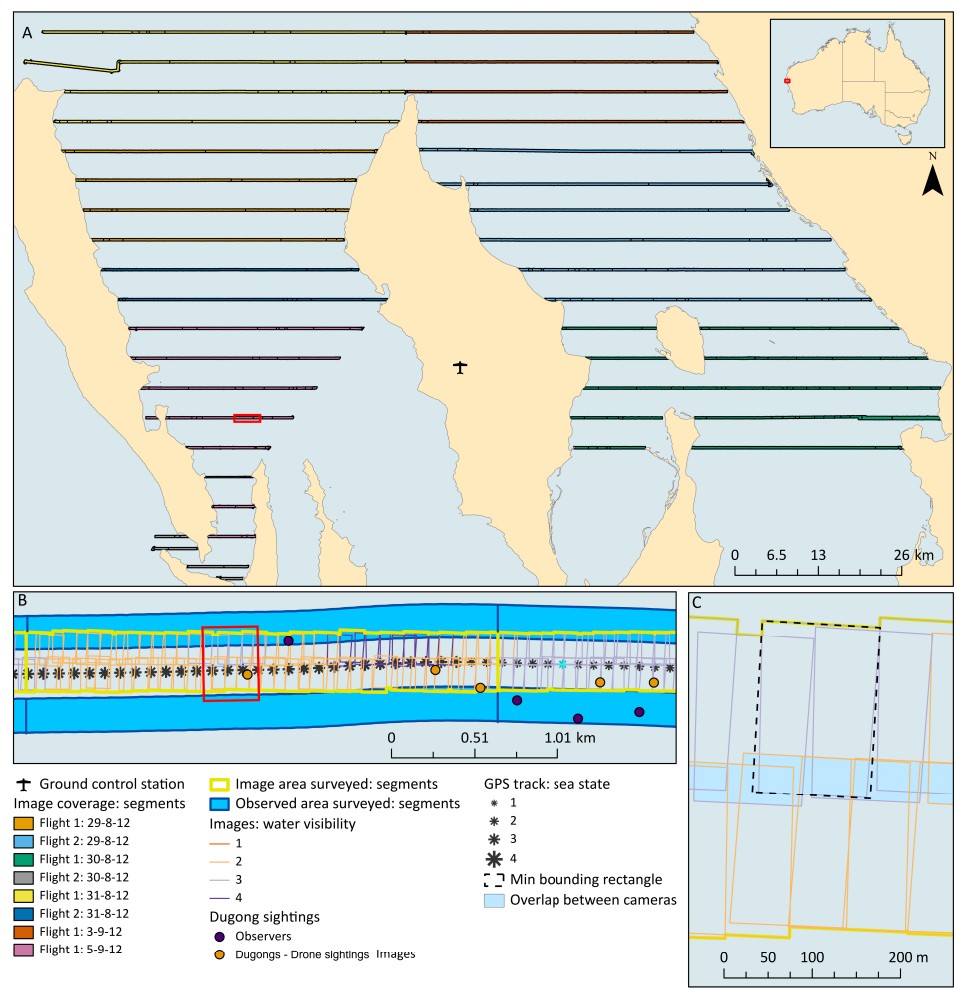

**Figure 2** **Maps showing study site, transects, image coverage, example transect segments and example image size and overlap.** Red extent indicators show the extent of the map of subsequent resolutions. (A) Coverage of images along each transect (split into segments) for each flight. Note that the area covered by flight 1, 29th was also covered within flight 2, 31st, and the area covered by flight 1, 5th was also covered within flight 2, 30th. (B) Example of the coverage from each platform along a portion of a transect, showing the splitting of the transect into segments of the same length for each platform according to water visibility scores from the images. The GPS track is shown with each point given a sea state score, which were averaged within each segment. Also showing dugong sightings mapped accurately within the images and coarsely from the manned sightings. (C) Closer view of images mapped and the calculation of the total area surveyed, side-lap between the cameras, and the on-ground width of images according to a minimum bounding rectangle.

was conducted under the Western Australia Department of Biodiversity Conservation and Attractions permits SF008415 and CE003616.

## Observers in piloted aircraft

The observer survey methods followed those described in *Marsh & Sinclair (1989b)*. We used a twin-engine Partenavia 68B flying at 100 knots and at an altitude of 500 ft (152 m). The survey team consisted of the pilot, survey leader, and four observers (two on each side
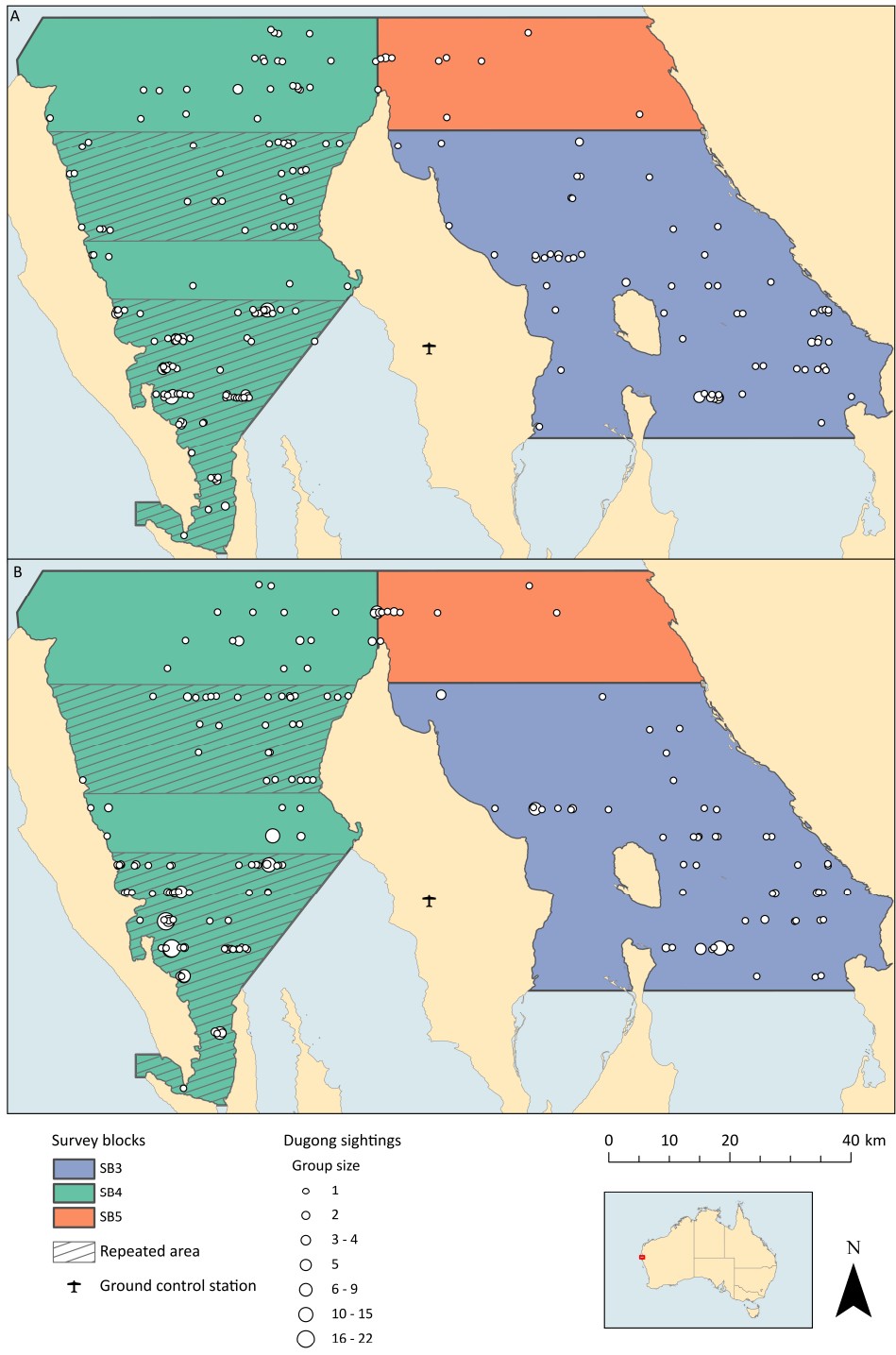

**Figure 3  Locations of dugong groups sighted throughout all flights by both platforms.** (A) Dugong groups sighted by the observers. (B) Dugong groups sighted within the drones images. Survey flights over the two hashed areas in block 4 were repeated twice.

of the aircraft). The 'back-seat' and 'front-seat' observers were visually and acoustically isolated so that their sightings were assumed independent. Using this double-observer system maximised the probability of the observer team sighting dugongs and allowed us to calculate 'perception bias' for this survey method (see below; *Marsh & Sinclair, 1989a*).

The observers announced all dugongs within a ∼200 m strip of sea on each side of the aircraft defined by rods attached to 'pseudo wing struts' (actual strip widths were 206 m on the port side and 203 m on the starboard side). The demarked transect strip was divided into 50 m zones (indicated by coloured tape along the pseudo wing struts), and observers called which zone the dugongs were sighted in, or whether the dugong was inside (below) or outside (above) the strip. These calls were recorded by a digital audio recorder, and the front-seat calls were transcribed in real-time onto a hand-held computer by the survey leader. The back-seat calls were transcribed post-survey and where there were discrepancies between calls on the same side of the aircraft, we assumed the front-seat observers' call was correct as they were the most experienced observers. Front- and back-seat calls of the same dugong groups were matched according to the timing of the call, the zone the sighting was in, and group size/composition. Given the limited transect strip and the moderate density of dugongs, these variables were sufficient to unambiguously match the independent sightings of the two observers on each side of the aircraft.

## Drone system

The drone survey flights were conducted using the *ScanEagle*, operated by Insitu Pacific Ltd, and described in detail in *Hodgson, Kelly & Peel (2013)* and *Hodgson, Peel & Kelly (2017)* as well as in Article S1 according to the *Barnas et al. (2020)* drone methods reporting protocol. Our flights were conducted over a remote marine environment, all-but eliminating the potential to capture people in the images or, therefore, impact privacy. The *ScanEagle* fixed-wing drone provides a relatively large range (up to 100 km, extendable using relay stations) and long endurance (24+ hrs) capabilities. The *ScanEagle* was operated *via* a Ground Control Station (GCS) which was a converted shipping container based at Monkey Mia airport (Fig. 2). We deployed the *ScanEagle* from the GCS using a pneumatic catapult "Superwedge" launcher and recovered it using the "Skyhook" retrieval system (*Hodgson, Kelly & Peel, 2013*; *Hodgson, Peel & Kelly, 2017*).

### Drone imaging system

The *ScanEagle* imaging system payload was an improved version of the custom payload described in *Hodgson, Kelly & Peel (2013)*. The aim of the new payload was to increase the swath width of the images (transect strip width) to match as closely as possible to the observer survey total strip width, whilst maintaining the ground sample distance (GSD—distance between pixel centres measured on the ground) of at least 3 cm shown to allow detection of dugongs (*Hodgson, Kelly & Peel, 2013*). We used two digital SLR cameras (24 megapixel (6,016 × 4,000) Nikon® D3200), each fitted with a standard 50 mm lens and a polarising filter set so that the direction of the polarisation was kept constant. According to pre-flight calculations, maximum coverage with minimal overlap was achieved by rotating each camera ∼11.5° from vertical in opposite directions (Fig. 4),

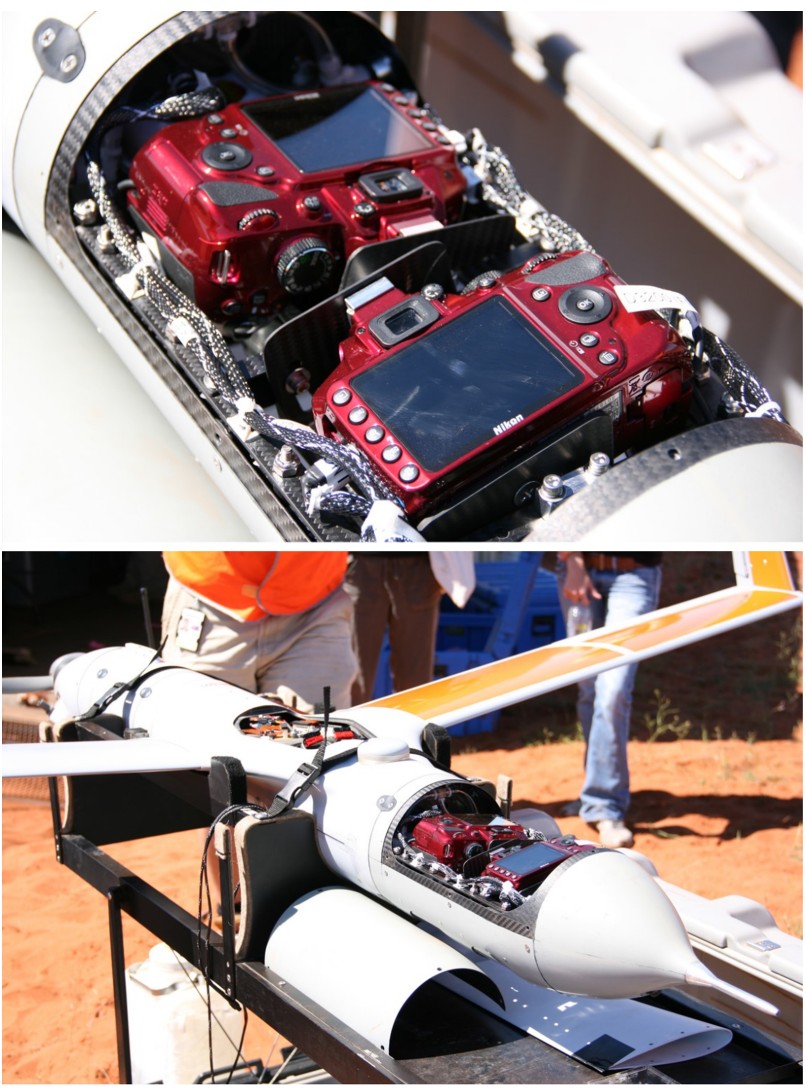

**Figure 4** Customised camera payload including two SLR cameras set at slightly oblique angles within the *ScanEagle* drone.

and our target of obtaining a 200 m width of coverage from each camera would be achieved by flying at 1,400 ft (427 m). We dropped the flight altitude to 1,300 ft (396 m) for some flights to increase the resolution slightly, as there was some concern upon initial review of the images about whether the dugongs could be confidently identified from the higher altitude (this concern was unfounded during the subsequent full review of images; see 'Results').

We had a fixed video camera in the nose of the *ScanEagle* from which imagery was viewed in real-time from the GCS, providing improved situational awareness. The SLR cameras were mounted within the airframe using a number of shock-absorbing mounts to reduce vibrations. Each image capture was tagged in real-time with GPS information from a dedicated receiver, as well as the drone's rotation information (pitch, azimuth and roll)
which was extracted for each image at the time of capture from the drone's telemetry data. Image capture was controlled (including start, stop and capture rate) *via* the GCS and the capture rate could be scheduled to achieve a prescribed proportion of image forward-lap (overlap between successive images) ensuring complete coverage along the transect lines flown. Our targeted image forward-lap was 40%, which was increased from the previous survey trial that showed the forward-lap of the images helped account for sightings being obscured by glare (*Hodgson, Kelly & Peel, 2013*). All images were stored on the camera's memory card and downloaded post flight.

## Coordination of the two aircraft

The drone was flown at an altitude of either 1,300 ft (396 m) or 1,400 ft (427 m) along the same transects at the same time as the Partenavia at 500 ft (152 m). The different altitudes of the two aircraft provided vertical separation in excess of that required by the Australian Civil Aviation Safety Authority and, together with regular communications between the two aircraft operators, ensured there was no safety risk to the Partenavia. The *ScanEagle's* ground speed ranged from 50 to 80 kn (target speed was 60 kn) and was therefore slower than that Partenavia. In order to ensure that each transect was flown by both aircraft as close in time as possible, long transects (>40 km) were split at the midway point. The Partenavia was flown in a holding pattern at that midway point to wait for the drone so that both started the second half of the transect at the same time. Similarly, the Partenavia's flight path between transects (*i.e.,* turn-around leg) was adjusted to allow the two aircraft to start each new transect at the same time. The timing was coordinated *via* radio whereby the drone operators informed the Partenavia pilot of their ETA for reaching the midway point or starting the next transect.

## Drone image review

All images were manually reviewed by one of three reviewers post survey. All reviewers used the same model computer (Dell 15'' 5540, FHD+ (1,920 × 1,080)), running Nikon's *ViewNX 2* software. The images are large (6,016 × 4,000 pixels), and our GSD was ~3 cm. In order for the dugongs to appear large enough on the screen to be detected easily, the images were reviewed at 50% of their actual size, and the reviewers used a standardised process to scroll around each image.

The reviewers scored each image with the following details:

- Number of dugongs
- Number of dugong calves
- Number classified (subjectively) as 'certain' and number 'uncertain'; uncertain sightings were either clearly fauna but of unclear taxa, or an 'animal shape' that could not confidently be distinguished within the image. Uncertain sightings were eliminated from the final dataset
- Number of double counts *i.e.,* the number of individual dugongs occurring within the forward-lap from the previous image (and therefore not new individuals)

Each image containing a sighting was then opened in a separate program (FastStone Image Viewer) and the sightings marked on the image for future reference, so (a) they

could be checked by an independent person, and (b) they could be used to train, test and validate our animal detection software (data not presented here). The reviewers were also asked to record the time they started and finished reviewing images in each continuous session, and the number of images reviewed, to provide an estimate of the manual image review rate.

## Perception bias
### *Observer surveys*

The human observer perception bias is the probability of a dugong group being seen by at least one observer from the team of two on each side of the aircraft, given the dugongs were available for detection. Perception bias was calculated as per *Pollock et al. (2006)* by fitting Huggins closed-capture models using the program MARK (*White & Burnham, 1999*) *via* the RMark package (*Laake, 2013*) in R (*R Core Team, 2022*). The perception probabilities used for each observer were those provided by the model that best fit the data according to Akaike's Information Criterion ($AIC_c$), which corrects for small sample bias. The perception probability could be the same for all observers, vary according to experience (primary or secondary observers), vary according to side of the aircraft (port or starboard), or be different for every observer. The probability that a dugong would be detected by at least one observer for each side of the aircraft was:

$$\hat{p}_d = 1 - (1 - \hat{p}_1)(1 - \hat{p}_2)$$

where $\hat{p}_1$ is the perception probability obtained for the primary and $\hat{p}_2$ the secondary observers ($i = 1,2$); the error for $\hat{p}_d$ estimate was derived using the delta method (*Oehlert, 1992*). For this analysis we used the data from all flights, regardless of repeats of the same areas (except the discarded flight as described in the Results).

### *Drone surveys*

In order to compare the manual detections of the three reviewers and calculate reviewer-wise perception bias, all were asked to review a particular set of images ($N = 3490$) from a drone flight conducted in the area where the highest density of dugongs occurred according to the observer survey. Using the capture history generated from detections within those images, perception probability was estimated also using a Huggins closed-capture model as described above for observers surveys, except that there were three reviewers and therefore two 'recaptures'.

We also tested the variation in detection of dugong groups in the aerial imagery arising from dugong group size and environmental conditions represented in the images (specifically water visibility and sun glitter; see below for further details on these variables). Group size was binned as 1, 2–3 and 4+. Several candidate models were considered to test combinations of the different sources in variation. These ranged from a constant model (where detection probability did not significantly vary between observers, group size, water visibility and sun glitter), through to a 'full' model containing interactions between observer and each of group size, water visibility and sun glitter. Model selection was based on Akaike's Information Criterion (AIC; *Burnham & Anderson, 2002*), but a version corrected for sample size ($AIC_C$).

We wanted to know whether our nominal number of images reviewed by all three reviewers was sufficient to assess perception bias, and to understand the minimum required for future surveys. However, rather than images, it is the number of detections that are relevant in understanding the precision of our detection probability estimates. We conducted a simulation exercise to determine the required minimum number of detections in the images when using two or three image reviewers (or this could be applied to a combination of reviewers and automated detection models; see 'Discussion'). Assumed detection probabilities for each reviewer was set, ranging from 0.4 to 1.0, going up in increments of 0.1. Total number of detections for each experiment ranged from 5 to 500, incrementing between 5 and 100 (increment lengths increasing as total number of detections increased). Each reviewer could have the same or different detection probability to the others in the scenario. In total there were 392 different combinations of detection probabilities and total number of detections for the two reviewers, and 2,744 for the three-reviewer scenario. Detection probability was estimated with a Huggins closed-capture model as described above (See Article S2 for details).

## Sampled areas and sighting locations
### Observer surveys

In order to better estimate the proportion of the survey area sampled by the observer team, we incorporated a correction for the actual altitude of the aircraft during the survey as recorded by the survey leader. We calculated an altitude fraction for each transect segment (see Water visibility, below, for an explanation of segments) as the average altitude recorded for that transect divided by the target altitude of 500 ft (152 m). The altitude fraction was then multiplied by the length of the transect segment and the combined width of the transect strips on either side of the aircraft (206 + 203 m) to get the effective area sampled for each transect segment.

In order to map the sightings as accurately as possible, in addition to transect 'zone' (see 'Observers in piloted aircraft') observers were also asked to call the time at which the dugongs sighted were abeam (perpendicular to the aircraft). The zone, 'time abeam', and GPS location of the aircraft at the 'time abeam' were used to map the observer dugong sightings at the centre of the zone in which it was called.

### Drone surveys

All images were imported into a custom version of the freeware, *VADAR* (Visual Acoustic Detection and Ranging; developed by Eric Kniest, University of Newcastle), which allowed us to map the true 'on-ground' footprint of each photograph, accounting for the *ScanEagle's* altitude, rotations (including the 'crabbing' effect caused by the heading of the *ScanEagle* being a few degrees off centre to compensate for crosswinds), and the orientation of the two cameras. Within *VADAR* we could also plot the position of each individual dugong marked within the image. An example is provided in Fig. 5, which shows sightings of multiple dugongs. *VADAR* enabled us to account for double-counted dugongs appearing in the side-lap (overlap of images) between the two cameras, as we were able to label each individual and link multiple sightings of the same animal. In Fig. 5, a number of dugongs

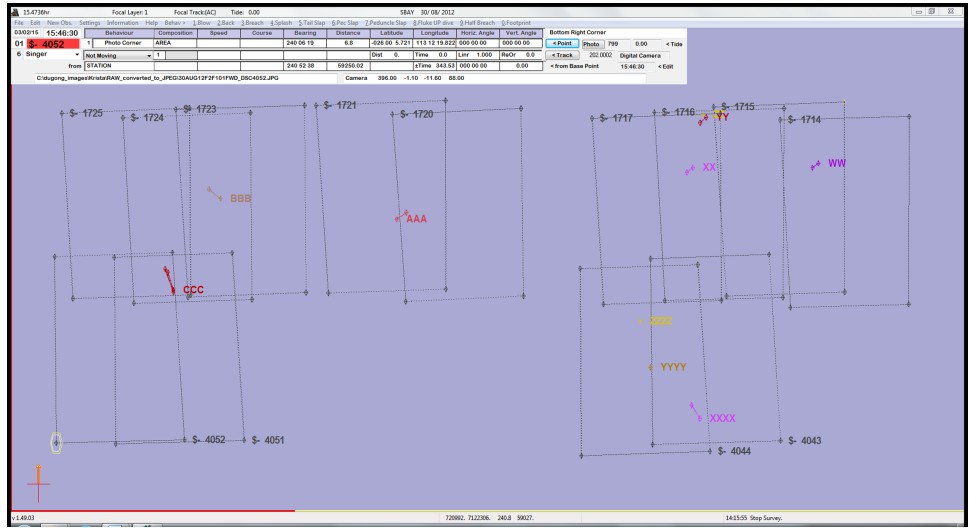

**Figure 5** **An example of images with sightings mapped in *VADAR*.** Images without sightings have been excluded for clarity. There are ten individual dugongs, each represented by letters and colours, with line links indicating individual dugongs appearing in multiple images.

appear in two images that overlap along the transect line (forward-lap) from one camera, while dugong 'CCC' appears in images from the two cameras (side-lap).

The data from *VADAR* was imported into ArcGIS (*ESRI, 2013*) so that we could calculate the total area surveyed by the drone. The four GPS locations of the corners of each image that were provided by VADAR were converted into a polygon, and we used the geoprocessing tool, 'Dissolve', to combine all images from each camera into two separate polygons for each flight. We then used the 'Merge' tool to combine the polygons from the two cameras. Total area per flight therefore represents the area within images regardless of whether captured in one or more images (*i.e.,* regardless of forward- or side-lap). We then used Editor to cut the surveyed area from each flight into transects and then transect segments, which allowed us to calculate the area per segment (see 'Water visibility').

We were also interested in determining whether we achieved the expected on-ground image width, forward-lap, and side-lap. To calculate image width, we used the 'Minimum Bounding Geometry' tool to get a bounding rectangle for each image. The longest side of this bounding rectangle represented the on-ground width of the image better than simply measuring along the edge of the image (Fig. 2). The forward-lap was confirmed by using the 'Intersect' tool to calculate the area of overlap and from that we calculated the percentage of overlap between each image. The percentage of side-lap was calculated by using the 'Intersect' tool to get the total area of overlap between the polygons created for each flight for each camera as described above.

## Environmental conditions
### Water visibility
The drone images were used to assess the water visibility of the areas surveyed by both the observers and drone imagery. One of the image reviewers subjectively scored each image

according to the predominant water visibility (which incorporated a measure of depth) according to the following categories:

1. Sea floor clearly visible (depth shallow)
2. Sea floor visible but obscured by turbidity (depth variable)
3. Sea floor not visible but clear water (depth likely > 5 m)
4. Sea floor not visible and turbid water (depth variable)

Both the observer and imagery transects were then manually segmented according to water visibility scores assigned to the drone images. These segments were of constant water visibility but varied in length (and therefore area surveyed; Fig. 2). All further analyses were then based on these segments.

### Sea state

The survey leader from the observer team recorded data on sea state according to the Beaufort scale. Sea state was recorded approximately every 2 mins, or when the conditions changed. A handheld GPS on board the aircraft provided a track of the observer survey flight path by recording a GPS location every second. Each of these GPS location points was assigned a sea state score according to the last record from the survey leader (Fig. 2). We then calculated the mean sea state for each transect segment as the mean score of all GPS track locations falling within that segment length.

### Sun glitter

Sun glitter was scored separately for each platform because the effect of this variable would have been different for each. The observer team recorded sun glitter for the north side of the aircraft (the direction from which sun glitter had the greatest effect) as a subjective estimate of the percentage area of the observers' view affected (0%, <25%, 25–50% and >50%). These scores were applied to the GPS track locations as described for sea state in order to get mean sun glitter for each observer survey transect segment.

To assess sun glitter within the drone images we used the same subjective assessment as for the observer surveys, except in this case one image reviewer estimated the proportion of each image affected by sun glitter. As per the observer survey assessment, we used only the images captured from the camera pointed north on each transect. The sun glitter score for each drone survey transect segment was the mean of all north-facing images in that segment.

### Cloud cover

Cloud cover was recorded in oktas by the observer survey leader at the beginning of each transect and applied to all observer and imagery segments in that transect.

## Comparing sightings between platforms

Overlap between the transect strips being observed during the observer flights, and the transect strip captured in the drone images was incomplete. For the observer surveys, the transect strips demarked by rods attached to each side of the aircraft had a gap in between, directly under the aircraft, that equated ~296 m at ground level when flying at 500 ft (152 m). The *ScanEagle*, however, was capturing images directly beneath the drone, and

therefore the transect strip of the drone was effectively in the gap between the observer strips, with some overlap on each side (Fig. 2). Our aim, however, was not to directly match the sightings from each platform, but to compare the aerial sighting rates obtained under the exact same conditions, *i.e.,* within the transect segments. This comparison is based on the assumption that the spatial distribution of animals did not systematically vary between the observer survey strips and the strips covered by the drone images.

We acknowledge that there may have been a slight difference between the observation windows afforded by each platform. Considering the narrow transect strip width, the observation window for the observers would have only been a few seconds and is usually considered instantaneous (*Pollock et al., 2006*). The forward-lap in the images meant that dugongs were visible in up to two images, providing a similarly small observation window of 2–3 s. We have therefore assumed the difference in observation windows to be negligible.

### Dugong counts

For each transect segment (the sample unit for animal counts), we tallied the total number of dugongs sighted by the observers and in the drone imagery (*i.e.,* the two platforms, excluding all double counts from both platforms), with each segment being accompanied by the associated covariates: water visibility (treated as categorical), sea state, observer sun glitter, imagery sun glitter and cloud cover. Exploratory analysis found no significant co-linearity between these covariates, and so these were treated as independent during model selection.

We fit a generalized linear mixed-effects model (GLMM) to the number of dugongs detected by the observer or imagery platforms to analyse the relationship between the dugong counts, platform and our environmental covariates. This analysis generally followed that described in *Hodgson, Kelly & Peel (2013)*. The response variable—dugong count, or the number of dugongs per segment—is considered to be Tweedie distributed in order to account for dugongs forming groups (*Williams et al., 2011*). The R package cplm (*Zhang, 2012*) was used to fit the Tweedie GLMMs.

As per *Hodgson, Kelly & Peel (2013)*, to address temporal autocorrelation in the number of dugongs in the survey area at any given time, we treated each flight as a random effect. Similarly, to address spatial autocorrelation within each flight, we treated each transect as another random effect nested within the respective flight. We fitted each random effect as an intercept only (assuming no interaction between the random effect and any of the fixed effects). We assumed compound symmetry in correlations within each level of the random effects, meaning that we assumed similar correlations among all images within the same level.

To account for the different strip widths, and therefore different areas, surveyed by each platform, we set the transect segment areas for each platform as an offset term.

The main effects were platform, water visibility (treated as categorical), sea state, sun glitter and cloud cover (which was binned into binary categories of cloud and no cloud), and we used backwards selection to eliminate terms in the models that were not significant.
### Number of groups and group sizes

In the observer survey, groups were defined subjectively by the observers as they were calling them, and usually a group of dugongs consisted of sightings that were made too close in time to be called separately. This means that dugongs that were spread perpendicular to the transect line (*e.g.*, right across the 200 m transect strip) were likely called as a group because they entered the observers view all at the same time, however, dugongs spread parallel to the transect line may have been called separately or as a group, depending on when the observer saw them.

For the imagery dugong sightings however, each individual dugong was georeferenced, and therefore a more rigorous definition of a group could be applied. For the purposes of comparison with the observer sightings, we defined a dugong group as being within 200 m of one other according to the chain rule, *i.e.,* as long as a dugong was within 200 m of its nearest neighbour, it was in the group.

For consistency, we reviewed the distances between observer survey dugong groups (estimated from their plotted locations as described in "Sampled areas and sighting locations") and merged groups that appeared to be less than 200 m apart, again according to the chain rule.

We examined the relationship between (a) the number of dugong groups seen per segment (group sighting rate—segment is the sample unit) and (b) group sizes (each group sighted was a sampling unit), and the main effects (platform and our environmental covariates) by fitting GLMMs as for dugong counts, using the same random effects and transect segment areas as offset terms. A Tweedie distribution was assumed for the group sighting rate response, but as the group size response is always >0 we used a Zero Truncated Poisson distribution.

## RESULTS

### Flight and coverage details

Observer aerial surveys were flown simultaneously with the *ScanEagle* drone in Shark Bay, WA, between 29 Aug and 5 Sept 2012. We conducted one training flight and nine survey flights over seven days. Some areas were flown more than once as we targeted areas of highest dugong densities (Figs. 2 and 3). Unfortunately, we had to discard the images from one flight because one camera did not capture properly.

The duration of each flight was limited by the observer aircraft, which could only fly a maximum of three hours according to fuel limitations. We extended some of the *ScanEagle* flights beyond the maximum endurance of the observer aircraft to capture extra data (not reported here), so our maximum duration *ScanEagle* flight was 4 hrs 14 mins.

During the eight survey flights, the two platforms each flew a total transect line distance of 1,602 km while 'on transect' (including repeated transects, Table 1). The on-ground coverage provided by the cameras from the *ScanEagle*, and the forward-lap between successive images along the transect, and side-lap between the two cameras, is depicted in Fig. 2. The observer team surveyed an estimated total of 645.3 km$^2$ across all flights, and the drone images covered a total of 576.7 km$^2$. Table 1 shows the number of transect segments

**Table 1  Details of flights, coverage and dugongs sighted from each platform (where Obs = observers and DI = drone imagery).**

| Flight | Drone altitude (ft/m) | Total length of transects (km) | Drone image coverage details | | | Total transect segments | Area (km2) | | Dugong count | | Images with dugongs[b] |
|---|---|---|---|---|---|---|---|---|---|---|---|
| | | | Mean (SD) % forward-lap | Mean width (m) | % side-lap[a] | | Obs | DI | Obs | DI | |
| 29F1 | 1,400/427 | 157.5 | 39.5 (9.3) | 210.2 (1.9) | 13.8 | 25 | 64.4 | 60.0 | 16 | 19 | 35 |
| 29F2 | 1,400/427 | 281.9 | 40.7 (8.1) | 208.7 (1.5) | 13.4 | 59 | 114.8 | 107.5 | 32 | 33 | 46 |
| 30F1 | 1,300/396 | 241.9 | 39 (11.2) | 199.7 (24.2) | 13.0 | 61 | 90.5 | 81.4 | 45 | 47 | 58 |
| 30F2 | 1,300/396 | 165.4 | 41.9 (9) | 193.4 (5.7) | 12.0 | 83 | 65.5 | 56.6 | 88 | 116 | 109 |
| 31F1 | 1,400/427 | 206.0 | 41 (8.1) | 206.2 (6.6) | 13.4 | 36 | 83.3 | 76.6 | 27 | 29 | 41 |
| 31F2 | 1,300/396 | 231.9 | 41 (9.1) | 193.9 (3.5) | 13.1 | 39 | 94.9 | 82.1 | 23 | 35 | 41 |
| 3F1 | 1,300/396 | 176.7 | 41.6 (9.3) | 194.4 (1.6) | 12.2 | 22 | 73.8 | 62.7 | 11 | 11 | 16 |
| 5F1 | 1,300/396 | 140.5 | 41.5 (8.8) | 193.9 (1.9) | 12.3 | 55 | 57.2 | 49.0 | 40 | 62 | 61 |
| Total | | 1,601.9 | | | | 382 | 645.3 | 576.7 | 282 | 352 | 407 |

**Notes.**
[a] This percentage overlap is based on the total area of coverage from each camera and the total area of overlap between the two.
[b] Regardless of double counts.

surveyed in each flight (with each segment representing constant water visibility), and the area surveyed by each platform per flight.

The forward-lap of successive images from a single camera along the transect was close to the prescribed 40%, and the on-ground width of images from each camera ranged from 193.4 m (captured at 1,300 ft, GSD = 3.2 cm) to 210.2 m (captured at 1,400 ft, GSD = 3.5 cm), which was close to our target of 200 m (Table 1). With the cameras angled obliquely at ∼11.5°, the side-lap between the two cameras along the transect line ranged 12–13.8%.

During Flight 2 on 31 August, the *ScanEagle* lost contact with the GCS while surveying the second most north-western transect. The *ScanEagle* is preprogramed to return to the GCS when this situation arises, but just after it started heading to base we regained communications and sent the *ScanEagle* back to its original flight path (*i.e.,* back on transect), and continued the remainder of the flight without incident (see Fig. 2). As the *ScanEagle* had only gone slightly off track and continued capturing images, we included this section in the analysis.

## Image review

A total of 44,694 images were captured from the two cameras on board the *ScanEagle* whilst 'on-effort', all of which were manually reviewed. The average time to review each image was 36 s. Therefore, the total time taken for a single person to review all images was in excess of 445 hrs. These estimates include the time taken to mark dugongs on the images for future reference. We note that reviewers could not conduct a full workday (*e.g.,* 7.5 h) of reviewing images as they needed many breaks.

**Table 2 Summary of environmental conditions recorded during each flight.** Means, minimums and maximums are based on the values calculated for the transect segments within each flight.

| Flight | Water visibility | | | Sea state | | | Cloud cover | | | Observer sun glitter north | | | Drone imagery sun glitter north | | |
|---|---|---|---|---|---|---|---|---|---|---|---|---|---|---|---|
| | Mean | Min | Max | Mean | Min | Max | Mean | Min | Max | Mean | Min | Max | Mean | Min | Max |
| 29F1 | 2.12 | 1 | 3 | 3.49 | 1 | 5 | 0.48 | 0 | 1 | 2.89 | 2 | 3 | 1.98 | 0 | 3 |
| 29F2 | 2.05 | 1 | 3 | 2.49 | 1 | 5 | 0 | 0 | 0 | 2.30 | 0.68 | 3 | 1.68 | 0 | 3 |
| 30F1 | 1.49 | 1 | 3 | 1.82 | 1 | 3.33 | 0 | 0 | 0 | 1.86 | 1 | 3 | 0.30 | 0 | 1.83 |
| 30F2 | 2.07 | 1 | 3 | 1.23 | 0 | 3 | 0 | 0 | 0 | 1.39 | 0 | 2 | 0.47 | 0 | 3 |
| 31F1 | 2.39 | 1 | 3 | 1.73 | 1 | 3 | 0 | 0 | 0 | 2.11 | 1 | 3 | 0.55 | 0 | 1 |
| 31F2 | 2.00 | 1 | 3 | 2.23 | 1 | 3 | 0.59 | 0 | 1 | 2.05 | 1 | 3 | 1.66 | 0 | 3 |
| 3F1 | 2.00 | 1 | 3 | 1.23 | 0.28 | 3 | 8 | 8 | 8 | 2.66 | 1 | 3 | 0.00 | 0 | 0 |
| 5F1 | 2.15 | 1 | 3 | 2.25 | 1 | 4 | 6.60 | 5 | 7 | 2.73 | 1 | 3 | 1.72 | 0 | 3 |

## Environmental conditions

The environmental conditions recorded during each flight are summarised in Table 2. Water visibility ranged 1–3 for all flights, with none of the areas surveyed considered to be water visibility 4—sea floor not visible and turbid waters; depth variable. We flew in relatively high winds during the first two flights, with sea state reaching 5 during parts of these flights. Cloud cover was minimal during all but the last two flights, with only one flight conducted in full cloud. Both the observer and imagery sun glitter was highest during the first flight, as expected, because this flight also experienced the highest sea states. The sun glitter scores were generally higher for the observer team, however, this difference may simply reflect the difference between scoring sun glitter when looking out the window of a plane *versus* looking at a single image, and the relatively subjective method of scoring sun glitter for both platforms.

## Sightings

The observer team saw a total of 282 dugongs (excluding duplicates from the front- and back-seat observers, but including those seen during repeated flights of the high-density areas), while the total drone imagery dugong count was 352 (excluding duplicates within image overlap; Table 1, Fig. 3). All dugong counts from the images exclude double-counts from forward- and side-lap. The total number of images captured that contained dugongs (regardless of double counts) was 407. Both platforms only saw one calf each during all flights combined, and therefore we could not make any comparisons about the relative ability of each platform to discern calves.

Although we detected more dugongs in the images than the observers saw, there were more groups sighted by the observers ($N = 218$) than in the images ($N = 186$; Table 3). The distinction was that there were bigger groups sighted in the images compared to from the observer plane (Fig. 3).

## Perception bias

The probability of the observers sighting dugongs, given they were available, was based on a Huggins closed capture model of best fit which was the model with different probability

**Table 3  Number of groups and mean group sizes seen by each platform.**

| Flight | Observers | | Drone imagery | |
|---|---|---|---|---|
| | Number of groups | Mean group size (SD) | Number of groups | Mean group size (SD) |
| 29F1 | 16 | 1 (0) | 18 | 1.06 (0.24) |
| 29F2 | 28 | 1.14 (0.36) | 24 | 1.38 (1.17) |
| 30F1 | 34 | 1.32 (0.98) | 28 | 1.68 (2.18) |
| 30F2 | 47 | 1.87 (2.47) | 41 | 2.83 (4.84) |
| 31F1 | 25 | 1.08 (0.4) | 17 | 1.71 (1.72) |
| 31F2 | 23 | 1 (0) | 20 | 1.75 (2.9) |
| 3F1 | 10 | 1.1 (0.32) | 9 | 1.22 (0.44) |
| 5F1 | 35 | 1.14 (0.55) | 30 | 2.07 (3.41) |
| Total | 218 | 1.29 (1.28) | 187 | 1.88 (3.03) |

**Table 4  Perception bias for each of the three image reviewers.**

| Image reviewer | Probability of detection | CV |
|---|---|---|
| 1 | 0.917 | 0.031 |
| 2 | 0.979 | 0.015 |
| 3 | 0.802 | 0.051 |
| combined | 1.000 | 0.000 |

estimates for the primary (front-seat) and secondary (back-seat) observers. The primary observers actually had a slightly lower probability of sighting dugongs ($0.74 \pm 0.03$ SE) than the secondary observers ($0.81 \pm 0.03$ SE) though the difference between the two was marginal. The overall perception probability for the double-observer teams on each side of the aircraft was 0.95 (SE = 0.009).

For the triple-reviewer approach to estimating the probability of individual reviewers detecting dugong groups in aerial images, model-selection based on $AIC_C$ indicated the most parsimonious model was one where detection rates differed by reviewer but did not include any other covariates. (See Data S3 for full details of model selection.) The probability of the image reviewers detecting dugongs that were visible within the subset of 3,490 images that all three reviewers processed ranged from 0.80 to 0.98, and for this subset of images the combined probability was 1 (Table 4).

Article S2 provides the results of our simulation to determine minimum detections required to estimate the probability of an image reviewer detecting available dugongs. This minimum depends on the desired level of precision. According to Fig. S2A, there is a rough asymptote point for the CV of the detection probability estimate, which equates to approximately 80–100 unique detections combined between the two observers (based on the fact that CV doesn't seem to decrease substantially after 100 detections, so insisting on more detections would run into the law of diminishing returns). Figure S2B shows that a similar number of detections ($\sim$75) would be required if estimating detection probability for three reviewers. Our image reviewers detected a combined total of 178 dugongs in the

set of images reviewed by all three to assess perception bias, which suggests our detection probability estimates had levels of precision which would be difficult to improve upon with increasing numbers of detections.

## Comparison between platforms
### Dugong counts
Our GLMM model suggested that platform (drone imagery or observer), water visibility and sea state all significantly affected the sighting rates of individual dugongs per segment (Fig. 6, Data S4). However, there was no interaction between platform and the environmental conditions. On average, the dugong counts from the drone imagery were 134% (95% CI [98–184]) of those from the observers.

For both platforms, dugong counts were significantly higher in our water visibility category 2 (sea floor visible but obscured by turbidity; depth variable) than in category 1 (sea floor clearly visible; depth shallow), with the category 2 counts being 154% (95% CI [99–240] of the dugong counts category 1. In turn, the dugong counts in visibility category 3 (sea floor not visible but clear water; depth likely > 5 m) were significantly lower than category 1, where the counts for the former were 47% (95% CI [28–80]) of the latter (Fig. 6, Data S4).

The dugong counts for both platforms decreased significantly as Beaufort sea state increased and this decline was almost linear (Fig. 6).

### Group counts
The only statistically significantly term remaining in our GLMM model describing the number of dugong groups counted per segment was water visibility, and the effects were similar to dugong counts (Fig. 7, Data S4). More groups were sighted in category 2 than in category 1 (134% (95% CI [90–198]) of those detected in the latter were detected in the former), while the number of groups per segment in category 3 was 54% (95% CI [35–85]) of category 1. Therefore, again, the relatively shallow water segments where the bottom was visible but obscured by turbidity had the highest group sighting rate.

### Group size
Our final GLMM describing group size included an interaction term for cloud cover and platform, along with Beaufort sea state and water visibility, with the latter two affecting both platforms equally. Group sizes decreased for both platforms as cloud cover increased, but the observer group sizes dropped much more dramatically—by 71% (95% CI [31–88]) compared to no cloud—than the group sizes detected in the drone images which only dropped by 14% (95% CI [−28–57]). The group sizes observed in the drone images were 224% (95% CI [167–302]) of those detected by observers when there was cloud, and when there was no cloud, group sizes in the images were 662% (95% CI [108–1218]) of those seen by observers (Fig. 8, Data S4).

The effects of water visibility were the same as for individual and group counts; the group sizes in category 2 (sea floor visible but obscured by turbidity) were 177% (95% CI [116–272]) of those in category 1 (sea floor clearly visible), and group sizes in category 3 (sea floor not visible but clear water) were 63% (95% CI [34–118]) of the sizes in category

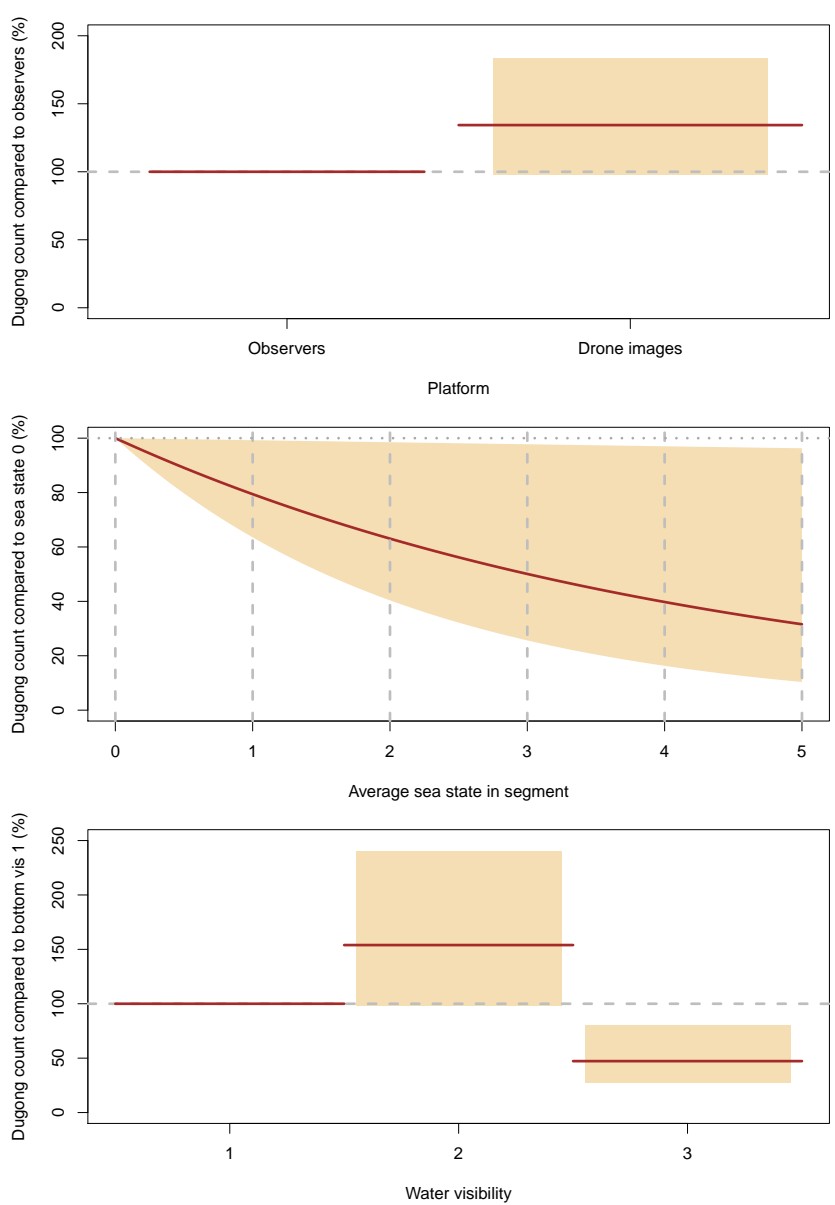

**Figure 6 Significant effects for dugong counts.** Expected (red line) and 95% confidence interval (coloured boxes/area).

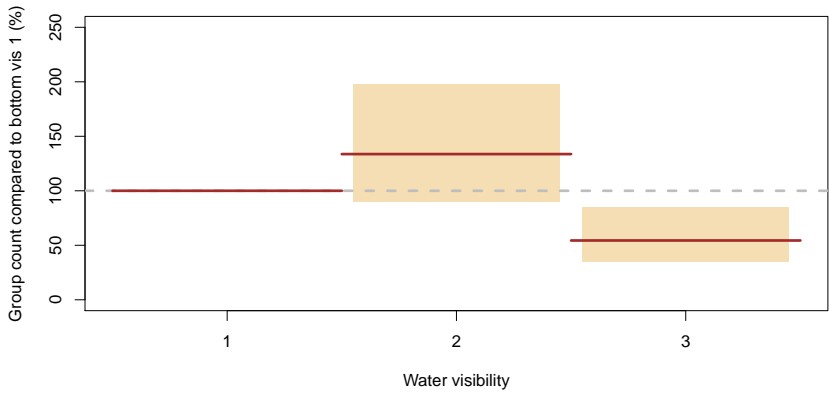

**Figure 7** **Significant effect for group counts.** Expected (red line) and 95% confidence interval (coloured boxes).

1. The effects of sea state on group size were similar to the effects on individual dugong counts; group sizes decreased as sea state worsened with group sizes in sea state 5 being 31% of those detected in sea state 0 (Fig. 8).

## DISCUSSION

### Comparison of sighting rates

This trial provided a direct comparison between sighting rates of dugongs by observers on board a piloted aircraft, and within images captured from the *ScanEagle* drone. Overall, the dugong sighting rate (*i.e.,* count of individual dugongs) was 1.3 (95% CI [0.98–1.84]) times higher from the drone images than from the observer survey. The group sighting rate was similar for the two platforms, however the group sizes detected within the drone images were significantly larger than those recorded by the observers, which explained the overall difference in sighting rates.

Other direct comparisons between observer and imagery data for similar wildlife surveys (*i.e.,* in-water marine fauna) have shown varied outcomes, with most suggesting observer and imagery provide comparative data. *Koski et al. (2013)* found no significant difference in large whale counts detected by observers and within imagery collected from the same aircraft during a survey of marine mammals over the Chukchi Sea, but detected higher rates of seals within the images. *Ferguson et al. (2018)* conducted a comparison of large-whale survey outcomes from three observation platforms: drone imagery, observers, and a camera mounted on the observers' aircraft. The surveys were conducted during the same time period (though not simultaneously) over the Chukchi and Beaufort Seas. They found that estimated densities of bowhead whales (*Balaena mysticetus*) were similar from the observers and the drone, and lower from the mounted camera, and that density estimates of beluga whales (*Delphinapterus leucas*) were highest from observer data. *Bröker et al. (2019)* directly compared narwhal (*Monodon monoceros*) sightings from observers and aerial imagery collected concurrently from a single aircraft, with both platforms producing similar numbers of individual animals and abundance estimates. *Garcia-Garin et al. (2020)*

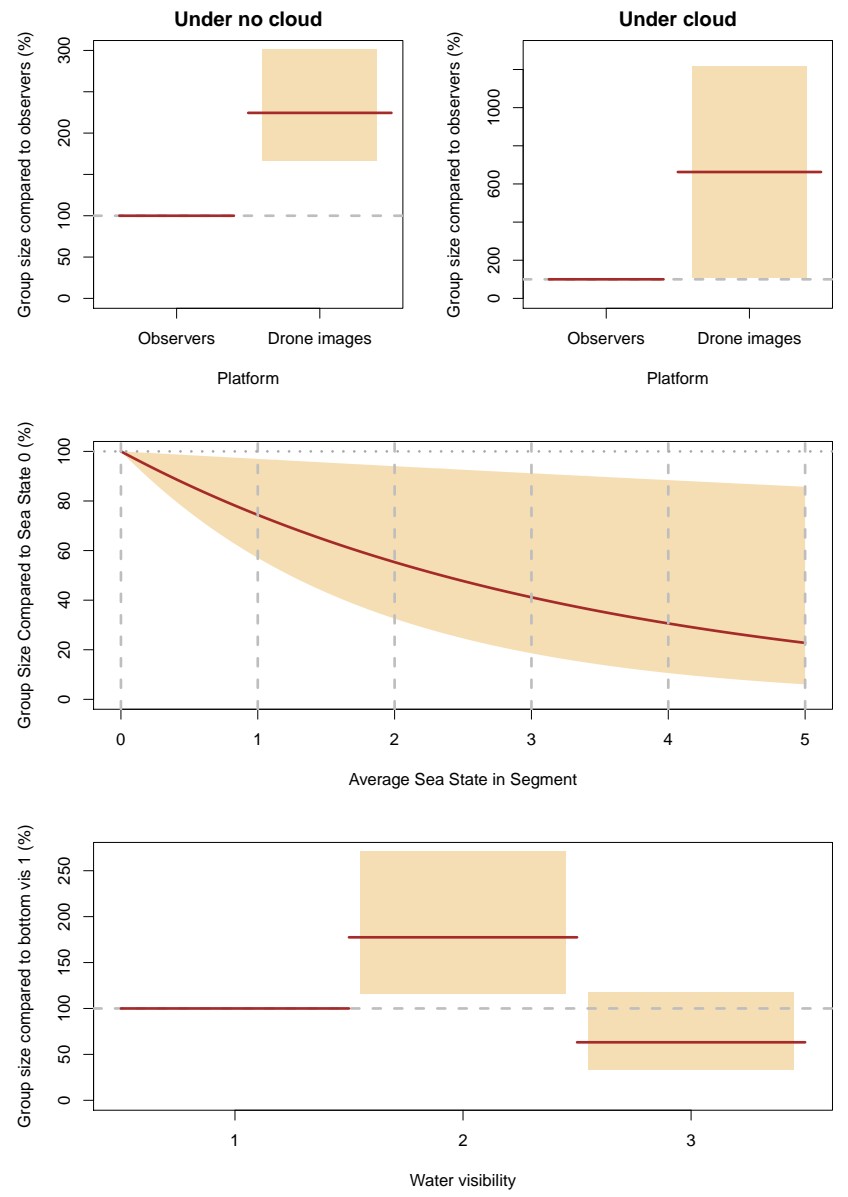

**Figure 8** **Significant effect for group size.** Expected (red line) and 95% confidence interval (coloured boxes/area).

similarly made a direct comparison between sightings of marine megafauna from observers and imagery from the same aircraft. They did not detect a significant difference in the total densities of all megafauna, though they note their low sample size may have contributed to this result.

## Effects of environmental conditions

Our investigation into the influence of environmental covariates on the sighting rates from observers *versus* drone imagery allowed us to determine whether these covariates could

explain the differences in detections between the two platforms. Cloud cover appeared to be the only covariate affecting the two platforms differently; the incidence of cloud cover resulted in smaller group sizes being detected by both platforms, but the observer group sizes dropped much more dramatically (by 71% (95% CI [31–88] compared to no cloud) than the group sizes detected in the drone images (14% (95% CI [−28–57])). We interpret this result with caution, as the majority of our flights were conducted in zero cloud (Table 2). However, *Aniceto et al. (2018)* and *Benavides, Fodrie & Johnston (2020)* also found that sighting rates were positively affected by increasing luminosity or light availability (during low cloud and maximum solar altitude) within images collected during small drone surveys of marine mammals and shark decoys respectively. Contrastingly, *Hodgson, Peel & Kelly (2017)* found that humpback whale sighting rates were not affected by cloud cover.

*Hodgson, Kelly & Peel (2013)*; *Hodgson, Peel & Kelly (2017)* found that dugong and humpback whale sighting rates were not affected by sea state or sun glitter. These two studies, along with the results presented here, suggest that the effects of sun-glitter obscuring portions of the images can be compensated by overlap between successive images along the transect line (forward-lap). Forward-lap also provides multiple detections of individual animals, which assists in identifying taxa. The lack of effect of sea state on sighting rates during those studies suggested that drone surveys could be conducted in a wider range of wind conditions than observer surveys. However, *Aniceto et al. (2018)* found that 'certainty of detection' (which was used as a proxy for detectability) for humpback and killer whales was negatively affected by increasing sea state. *Benavides, Fodrie & Johnston (2020)* also showed that shark decoys were less likely to be sighted in drone images when images were captured in winds ≥ 8.2 knots. *Hodgson, Kelly & Peel (2013)*; *Hodgson, Peel & Kelly (2017)* may not have had large enough sample sizes from the various sea state conditions to adequately test the effects of high sea states. In our current study, sea state did affect dugong counts and group size estimates which decreased as sea state worsened (an effect observed for both the observer and imagery platforms). Future studies should perhaps investigate what level of this effect is acceptable. During our study, even at Beaufort sea state 2, the sighting rate of dugongs dropped by 37% (95% CI [2–60]; Fig. 6), and this is usually considered suitable conditions for small marine mammal observer surveys (*e.g.*, *Pollock et al., (2006)*) considered Beaufort sea state ≤ 2 as optimal conditions.

Similar to the findings by *Hodgson, Kelly & Peel (2013)*, water visibility consistently affected the sighting rates according to the three variables tested (individual counts, group counts and group size). Both observers and the drone imagery were significantly more likely to detect dugongs in areas where the sea floor was visible but obscured by turbidity than in shallow water areas where the sea floor was clearly visible or where the water was relatively clear with the bottom not visible (*i.e.*, relatively deep water). We suggest that these results reflect dugong and/or seagrass distribution within our study area rather than the 'detectability' of the dugongs in each of these environments.

## Differences in group sizes

The difficulties that observers have in sighting dugongs in large groups was noted by *Marsh & Sinclair (1989b)*, who subsequently developed a protocol to break from transect,

circle, count and photograph any dugong groups of >10 individuals. Similar protocols are used for aerial surveys of small cetaceans (*Slooten, Dawson & Rayment, 2004*; *Bilgmann, Parra & Möller, 2018*). This protocol, however, is dependent on observers seeing at least 10 animals during the initial pass. As groups can be relatively spread out (we defined a group using a 200 m chain rule) it is possible that while observers are detecting some animals, they are missing others that are nearby. According to our analysis, dugongs were much more likely to be in areas where the bottom visibly was 'sea floor visible but obscured by turbidity; depth variable' (category 2). During our surveys the observers noted that in this water visibility category dugongs were quite difficult to see—they would initially see one or two dugongs, and then upon searching more, realise they could see more animals. However, they only had a matter of seconds to determine if >10 animals were present and initiate a circle-back. Similarly, there were a number of drone images where the dugongs were barely visible and it took some time of continual scrutiny of an image before being confident that all visible dugongs had been detected. Often many more dugongs were visible in the image than would have been detected in the first few seconds of review. The larger group sizes seen within the images may therefore be explained by the extra time that reviewers can take to detect dugongs compared to observers making calls in real-time. If this hypothesis is true, it suggests that perception bias may often be underestimated for observer surveys, even when using two observers on each side of the plane; more work is required to determine if this represents a level of unmodelled heterogeneity in the detection process that needs to be accounted for. This effect appears to have been amplified by cloud cover, with cloud affecting the observers' ability to detect large groups more dramatically than reviewers of the drones images. Cloud cover has not previously been considered a factor affecting dugong sighting rates during observer surveys and is not normally corrected for in estimating population abundance (*Pollock et al., 2006*).

Another possible factor contributing to the difference in group sizes between the two platforms is the difference in the viewing area. While the images from the two cameras covered one full transect strip of approximately 400 m, the observers were calling sightings within two separate 200 m strips on either side of the aircraft. We used a relatively large distance for defining groups (*i.e.,* the 200 m chain rule). Large groups are more likely to be near the edge of the transect strips we used for the observer aircraft, than for the drone images, and therefore more likely to be partially outside the strip and not counted in their entirety. Therefore, the split transect strips covered by the observer platform would be expected to have a smaller group size, than the continuous transect strip covered by the drone. This theory warrants further investigation to determine whether strip width/coverage by cameras might ultimately affect population estimates.

## Comparing historical observer data with imagery data

The main aim of our trial surveys was to determine whether drones could replace observer surveys and provide data that can be compared to historical data. Our trial survey was the first to suggest that drone image surveys of in-water marine fauna can provide superior data over human observers. We note this result may be specific to drones; previous studies have not always detected a difference between sighting rates from observers and imagery from

piloted aircraft (*Koski et al., 2013*; *Ferguson et al., 2018*; *Bröker et al., 2019*; *Garcia-Garin et al., 2020*). Nonetheless, if there are differences in the data being collected from drone images compared to historical observer data, what does this mean for transitioning to drone or imagery surveys?

In Fig. 1, we consider the variables that may affect detection probability for the two survey platforms. For availability bias, both platforms will likely be affected by the same set of variables, and according to our data, the only variable that affects detection from the two platforms differently is cloud cover. Therefore, the observer-based corrections for sea state and water visibility given by *Pollock et al. (2006)* and *Hagihara et al. (2018)* can theoretically be applied to drone imagery data, and separate corrections for cloud cover may need to be developed for both observer data and imagery data.

We then consider the variables that can affect perception bias, and in the processing of imagery, we consider both manual detections and automated detections from deep learning models (Fig. 1). Surveys using double-observer teams report a relatively low perception bias. Although the probability of individual observers can vary significantly (*e.g.*, in *Pollock et al. (2006)* probabilities ranged from 0.24 to 0.9) the overall probability of a tandem team seeing a dugong is generally quite high (in the Pollock et al. example, probabilities ranged from 0.87 to 0.96, and similarly, *Parra et al. (2021)* and *Slooten, Dawson & Rayment (2004)* report detection rates of small dolphins by double-observer teams as 0.91–0.99 and 0.96 respectively. Both sets of observers from our observer survey team were estimated to have seen 95% of dugongs available to be detected.

Our image reviewer detection rates varied between 0.8 and 0.98, and the variation in our case may have been a result of the experience of the reviewers; the reviewer who had done the most observer surveys had the highest detection rate, while the reviewer with the lowest rate had never done an observer survey. *Odzer et al. (2022)* similarly found that their image reviewer with the most experience conducting observer surveys detected the most turtle decoys in their drone images, and that the probability of three reviewers of mixed experience detecting turtles was not significantly greater than that of a single experienced reviewer. This variation suggests it is important to quantify and account for perception bias by having multiple reviewers process at least a subset of images. The proportion of images that need to be reviewed by multiple reviewers depends on the sighting rate. Our simulations suggest that for two reviewers, at least 80 combined animal sightings are required and for three reviewers this could be dropped to 75 sightings, beyond which point the precision of the perception estimates will not improve substantially with increasing detection numbers. It should be noted that the number of detections that might be sought is potentially larger when including covariate effects in the double-observer analyses. We note that the combined detection rate for all three image reviewers was 1. Meaning that if all three reviewers reviewed all images, they would be almost certain to see all available groups. Using three reviewers for the entire image dataset would, however, significantly increase the time and cost of this process.

Our results are applicable to the imaging system, capture protocol and processing methods that we used. Changing any of these variables could affect the quality of images and the ability to detect animals in images, and therefore affect final population estimates.

Further studies are needed to investigate the effects of these image capture and processing variables on detections, and whether any variation caused by these factors significantly affects final estimates of abundance.

The widespread use of drones for wildlife surveys, is limited by the image processing required (*Christie et al., 2016*) and inevitably, these large image datasets will need to be processed using deep learning models to detect the target animals. However, detecting small marine fauna in images is a challenging computing task, particularly for animals like dugongs that are often detected on the sea floor which means there is a large amount of background 'noise' in the images. A deep learning model needs to discern marine fauna from various structures and patterns on the seabed, as well as white caps, sun glitter and debris on the water surface.

*Maire, Mejias & Hodgson (2015)* had success in producing an automated model to detect dugongs in our images using convolutional neural networks. The rate of positive detections is reported as 80%. This recall rate can be considered as a proxy for perception bias, however it is important to consider the potential error associated validating the detections produced by a computer model (Fig. 1), where again there is potential for true detections to be missed and false positives to be verified as true detections.

## Future implementation of drone surveys

Our trial surveys have shown that drone imagery provides comparable data to observer surveys of dugongs. To be confident in moving forward with imagery surveys, particularly in the case of dugongs for which there is a long-term data series from observer surveys, it would be useful to conduct some further assessments of the effects of water turbidity. Throughout our survey area the water clarity was relatively high. However, dugongs regularly occur in areas of high turbidity, so to complete the comparison between observer and imagery data, it would be prudent to conduct this experiment over turbid dugong habitat.

At the time we conducted this survey trial, the extent of the *ScanEagle's* range from our GCS did not quite cover the extent of a standard full dugong survey of Shark Bay (see *Holley, Lawler & Gales, 2006*). However, this range could be extended by using repeaters to hand off control to a relay station (small ground vehicle). In other scenarios, multiple drones could be flown—themselves acting as repeaters.

High-end drones like the *ScanEagle* may not yet be affordable in the standard wildlife research and monitoring budget, particularly for a one-off survey. It is not appropriate for us to comment on the costs of our trial surveys as our drone operator provided in-kind support, and costs have likely changed (and will continue to change). However, drone models with similar range and endurance, but that are purpose designed and manufactured for survey, may prove more cost-effective. Where drones are logistically or financially unviable, it is possible to conduct imagery surveys using piloted aircraft, and as discussed above, a number of studies have assessed this method. Our comparison may not be directly transferable to a piloted aircraft imagery survey because of differences in flight speed, stability and operational factors. The comparability of drone and piloted aircraft imagery surveys needs further investigation.

The efficacy of imagery surveys will inevitably improve as we continue to understand the detection probability of our target species, and the implications of conducting imaging surveys using different platforms and image capture systems and methods. There is an opportunity to standardise survey data collection by automating much of the process of extracting data from the images, as well as to use deep learning models to further interrogate the factors affecting sighting rates. Here we have focused on replicating a standard large-scale observer survey using drones. However, as imagery surveys become more common-place, there is an opportunity to design surveys differently (*e.g.*, *Cleguer et al., 2021*) and replicate surveys with minimal variation in perception bias. Imagery surveys can also be designed and analysed remotely, providing the possibility of survey data to be captured by non-experts and in previously unsurveyed locations.

## CONCLUSIONS

We compared dugong sighting data from observers in piloted aircraft to those from drone imagery collected concurrently and found that larger dugong groups were sighted in the drone imagery than by observers, which meant that more dugongs were sighted in the drone imagery overall. Water visibility, Beaufort sea state and sun glitter all affected the sightings from both platforms to the same degree. However, cloud cover had a larger impact on the observer sightings than the drone imagery sightings—in the incidence of cloud cover, the sizes of dugong groups detected by the observers fell much more dramatically than those from the drone imagery.

If transitioning a long-term aerial survey monitoring program from observer to imagery surveys, our comparison of dugong sightings has suggested that the detection rates from the two platforms could be considered equivalent, apart from the size of groups than can be detected in the incidence of cloud cover. However, we caution that our results are limited to a study site with relatively good water visibility, and to imagery collected from drones rather than a piloted aircraft. Further comparisons would be needed to understand the implications of high water turbidity and capturing imagery from varying platforms and imaging systems.

## ACKNOWLEDGEMENTS

The *ScanEagle* drone was supplied, customised and operated by Insitu Pacific Ltd as in-kind support and we thank the IPL team—Michael Wilson, Nigel Meadows, Carl Brown, Tony Mountford, Peter Cassimatis, Lennon Cork, and particularly Neil Smith, for their dedication to this project. The survey observers were Jodie Mehrtens, Melinda Rekdahl, Verity Steptoe and Kasey Darts, and the survey leader was Elizabeth Burgess. We thank David Holley and the Department of Biodiversity, Conservation and Attractions (DBCA) for their in-kind support to this project. Krista Nicholson, Verity Steptoe and Kate Sprogis manually reviewed the images. We are grateful to Eric Kniest for customising VADAR for our aerial imagery and Krista Nicholson for georeferencing the imagery and sightings, and scoring the imagery for environmental conditions.

### Funding

This project was funded by the Australian Marine Mammal Centre (AMMC) under the Bill Dawbin Postdoctoral Research Award granted to Amanda Hodgson. Additional funding was received from AMMC to conduct the manual image review. The funders had no role in study design, data collection and analysis, decision to publish, or preparation of the manuscript.

### Grant Disclosures

The following grant information was disclosed by the authors:
Australian Marine Mammal Centre.

### Competing Interests

The authors declare there are no competing interests.

### Author Contributions

- Amanda J. Hodgson conceived and designed the experiments, performed the experiments, analyzed the data, prepared figures and/or tables, authored or reviewed drafts of the article, and approved the final draft.
- Nat Kelly conceived and designed the experiments, analyzed the data, prepared figures and/or tables, authored or reviewed drafts of the article, and approved the final draft.
- David Peel conceived and designed the experiments, analyzed the data, prepared figures and/or tables, authored or reviewed drafts of the article, and approved the final draft.

### Animal Ethics

The following information was supplied relating to ethical approvals (i.e., approving body and any reference numbers):

Murdoch University Animal Ethics Committee provided approval for this research under permit R2365/10.

### Field Study Permissions

The following information was supplied relating to field study approvals (i.e., approving body and any reference numbers):

This field work was approved by the Western Australia Department of Biodiversity Conservation and Attractions under permits SF008415 and CE003616. There was minimal potential for the drone operations to impact the privacy of people captured as imagery was captured over a remote marine environment (there were no boats captured in any of our images).

### Data Availability

The data is available on Edith Cowan University's Research Online Institutional Repository: Hodgson, A. (2023). Drones for large-scale wildlife surveys: Raw data to support manuscript - Hodgson et al. Edith Cowan University. https://doi.org/10.25958/fkdd-qb81.

## Supplemental Information

Supplemental information for this article can be found online at http://dx.doi.org/10.7717/peerj.16186#supplemental-information.

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
