# Peer review of "Drone images afford more detections of marine wildlife than real-time observers during simultaneous large-scale surveys"

_PeerJ, doi:10.7717/peerj.16186_

## Round 0.1 · original submission · Minor Revisions

Both reviewers appreciate the work you've put into this paper, and have highlighted a few areas that could be improved (e.g. additional comparisons suggested by reviewer 2). Reviewer 1 also flagged the availability of data, which I was also unable to access.

·

Basic reporting

This paper compared dugong sighting data from observers in piloted aircraft to those from drone collected concurrently and found that larger dugong groups were sighted in the drone imagery than by observers, which meant that more dugongs were sighted in the drone imagery overall. Water visibility, Beaufort sea state and sun glitter all affected the sightings from both platforms to the same degree. However, cloud cover had a larger impact on the observer sightings than the drone imagery sightings - in the incidence of cloud cover, the sizes of dugong groups detected by the observers fell much more dramatically than those from the drone imagery.

Overall this paper is self-contained and was well written in a professional manner and easy to follow. The literature references were sufficient. The figures and tables were great, easy to read and told the story well.

The raw data and processed image data were not provided. The code for the simulation was provided, which was helpful and well commented.

Experimental design

They clearly defined their question and why it was a useful question.

This paper fills in a knowledge gap and will be cited often in the future, I would predict.

They used appropriate experimental design, data collection methods, and analyzes, with an appropriate level of details.

The simulation work to investigate a sufficient sample size for the reviewer’s comparison was an excellent addition to support the conclusions.

The tables, figures, and supplementary appendices were all useful.

In SuppS3 Table 1, what is "time" in the models?

Validity of the findings

This work is unique and will be very helpful for other researchers entertaining the idea of collecting digital abundance survey data from drones or human piloted planes.

The conclusions were well stated and supported.

I don't think the underlying data were provided. I would assume it is not needed to provide the images, but maybe you could provide tables of the numbers of animals per image and per human observed. But I personally not sure this is necessary unless required by the journal.

However, in the title and the text I’m not sure the word “superior” is appropriate. The images did result in seeing more than the humans but the difference was not that large (even though it was statistically different). I personally like a word like “improved” or something less dramatic.

Again in the title and in the text I’m also not sure about the word “large-scale”. It was not a tiny study area, but it was not a large study area in my view. Not sure it makes much different, but …

Additional comments

no additional comments

·

Basic reporting

The manuscript is well written, introduces its context very clearly and cites all the most relevant references for this topic. The structure of the methods and results makes sense but the manuscript relies on a significant amount of Supplemental material, which forces the reader to do quite a bit of back and forth.

S1 repeats the context and overall objectives of the manuscript, as well as Figure 2 of the manuscript, which is probably unnecessary.

Experimental design

Although this study builds on previous ones by the same author(s) about the use of drones for marine mammal research, it presents original research and a novel contribution by aiming to answer specific questions about the comparability of detection rates in large scale surveys. The gaps in the literature are well identified and the need for this work well justified. This work is timely, as these new technologies become increasingly important in the field of wildlife surveys. As described in the introduction, an important issue with this technological transition is the importance of maintaining the validity of long time-series of abundance estimation and habitat use.

In terms of experimental design, the main value of this study is the simultaneous surveying by the two platforms, thus insuring the counts are influenced by the same environmental conditions (though possibly in different ways, as shown by the results), and that any difference in detection rates are not due to random variation in the distribution of animals or groups among the survey lines. Moreover, unlike previous studies, the drone in this survey was not restricted to flying within line of sight and therefore this experimental design does a better job of mimicking real survey conditions. These points alone constitute an important contribution and justify its publication. The manuscript also contains a useful review of factors affecting detection of marine mammals in aerial surveys and practical considerations on detection rates when reading drone imagery.

However, I wonder if the stated objectives of the analysis are the most useful that they could have been. The authors emphasize that comparability of time series of abundance estimates when switching to a new method is one of the main incentives for this work. But such time series would not contain only the raw number of counts or total individuals, they would report the actual abundance estimates taking into account covered area (i.e., density) and any correction factor applied to it (e.g., perception). This study does consider all these elements but chooses to focus its conclusions on the “sighting rates” along the transects without taking into account the area covered and the perception multipliers. I understand, as explained in lines 409-413, that the aim was not to directly match the sightings from each platform with one another, and I agree with that decision because ultimately what matters is the abundance estimate. But sighting rates alone (i.e., counts per segment or per km) are not necessarily comparable if the areas covered are different. For instance, if one was comparing aerial surveys taking photos at two different altitudes, with one altitude providing double the coverage, it would not make sense to compare the number of individuals detected without taking into account the area.

I think there should either be more emphasis in the reporting and analysis on the density of sightings, or better justification as to why sighting rates alone are considered important. (See also comment below on the interpretation of the results.)

Validity of the findings

Duplicate identification for perception bias analysis: Lines 264-266 and S3 provide explanation on the data and models used to estimate perception bias, which implies the identification of duplicate sightings between the front and back observers (to identify “recaptures”) but there is no description of the criteria used to identify these duplicates. For numerous marine mammals, this process is actually not trivial and relies on various rules-of-thumb and thresholds (e.g., proximity in time, in angle, in group size). How was this done in this strip survey?

Clarity on numbers: Lines 415-416 mention that the authors “tallied the total number of dugongs sighted by the observer” but it’s not clear if the numbers provided in line 521 represents the total number of sightings for both sets of observers, or the number of unique sightings (i.e., after removing duplicates between front and rear observers). Also, why is there 282 visual sightings in table 1 but 287 are mentioned in S2? Is that the difference between unique and total sightings? Finally, if one multiplies the 218 groups in table 3 by the mean group size of 1.22, the result is 266 individuals, which is not the same as the totals reported in table 1.

Interpretation of the results: As stated above, it would be interesting to compare not just the sighting rates but the actual density of individuals. Using the values in table 1 and dividing by the area covered, the density in the drone flight is actually 1.40 times higher than that of visual detections (a greater difference than that reported for sighting rates). Taking into account that this estimate should then be divided by the combined perception probability for the double-observer team (i.e., 0.952), whereas the drone-based estimate would remain unchanged because the combined reviewer detection rate given in S3 is 1 (something that should be added to line 547 and table 4), this would reduce the difference between the final estimates to a factor of 1.33. I feel that giving the readers the results for all of these steps in sequence (e.g., in a table), would allow them to see for themselves how the final abundance estimates would be affected (with different effects going in different directions).

Availability bias: The authors mention and explain availability bias in the introduction. In lines 728-734, they assume that both platforms in this study would have been affected by the same variables (i.e., dugong diving behaviour and visibility through the water would be the same for both methods) and therefore that the sighting rates can be compared without availability corrections. I think this warrants more discussion. I can think of two factors that might create a different availability bias between drones and observers: a) photo analysts have more time to discern submerged animals that may be at the deeper limit of visibility and therefore they might count more individuals in a group (indeed, this may be one of the reasons for the difference observed in group size); and b), visual surveys are often corrected for time-in-view, i.e., an instantaneous correction factor is applied to photographs but for visual observers this factor is modified for the time that animals could have been in their field of view, which can allow submerged animals to appear at the surface while the aircraft is passing over them, and therefore has the effect of reducing the bias and its correction factor. Here, such an adjustment for time-in-view would further exacerbate the difference between visual and drone estimates. Perhaps the time-in-view adjustment is negligible for dugong given their diving behaviour and the speed of the aircraft, but it should at least be mentioned.

Additional comments

Title: When I read the title “Drone images provide superior data over human observers…”, I expected that this statement was supported by a comparison with the known truth (e.g., a population of known size, or some simulated survey). What the results make clear, however, is that the counts are greater. I would argue that greater is not always superior (i.e., “better”). I admit that if more dugong are seen in the exact same transects, then it would suggest the method is superior because it’s missing less animals. However, an important aspect of abundance estimates is their precision. Without a full calculation of the abundance estimates (including all corrections) and their CVs, it is difficult to say which approach is superior on the basis of reliability and precision. The authors are also right to mention questions of cost and time in their discussion, as in some cases more frequent surveys might be better than fewer superior ones. It might very well be that drones eventually provide better estimates, but I don’t think the analysis presented here justifies the statement that “drone images provide superior data”. I would suggest either a more neutral title (e.g., Comparison of…, Comparability of counts…, etc.) or a more precise one (e.g., Drone images result in higher group sizes…).

---

## Round 0.2 · accepted · Accept

Thanks for making the changes requested by the reviewers, and I appreciate making the data available. Congratulations on your publication!